# Gold Nanocages with a Long Surface Plasmon Resonance Peak Wavelength as Contrast Agents for Optical Coherence Tomography Imaging at 1060 nm

**DOI:** 10.3390/nano15100755

**Published:** 2025-05-18

**Authors:** Yongping Chen, Jiefeng Xi, Vinh Nguyen Du Le, Jessica Ramella-Roman, Xingde Li

**Affiliations:** 1Department of Biomedical Engineering, Johns Hopkins University, Baltimore, MD 21205, USA; 2Noble Life Sciences, Sykesville, MD 21784, USA; 3Biomedical Engineering Department, Florida International University, Miami, FL 33174, USA; v.n.du.le@uah.edu (V.N.D.L.);

**Keywords:** gold nanocage, contrast agent, optical coherence tomography (OCT), biomedical imaging

## Abstract

There is growing interest in optical coherence tomography (OCT) imaging at a wavelength of 1060 nm. However, potential contrast agents for OCT imaging at this specific wavelength has not been thoroughly investigated. In this study, we present the synthesis and optical characterization of gold nanocages with a small edge length (~65 nm) and a surface plasmon resonance peak around 1060 nm. These nanocages represent a class of potential contrast agents for OCT at 1060 nm. OCT imaging experiments were conducted on phantoms and in vivo mouse tissues using a 1060 nm swept-source OCT system, demonstrating significant enhancement in imaging contrast due to the presence of the gold nanocages.

## 1. Introduction

Optical Coherence Tomography (OCT) is a noninvasive imaging technology that enables real-time, high-resolution visualization of biological tissues [1]. Swept-source OCT (SS-OCT) has emerged as a powerful advancement, offering fast A-line scan rates and high detection sensitivity. Conventional OCT imaging contrast is primarily governed by the intrinsic optical scattering and absorption properties of biological tissues (often dominated by scattering). Similar to other imaging modalities, such as X-ray computed tomography, ultrasound, and magnetic resonance imaging, the use of exogenous contrast agents can significantly enhance the imaging capabilities of OCT and, more importantly, molecular specificity. Contrast enhancement has been achieved through the use of various agents, including dyes [2], microspheres [3], microbubbles [4], gold superclusters [5], and gold nanoparticles, such as nanorods [6], gold superclusters, nanoshells [7,8], nanostars [9], and nanocages [10,11]. In addition to contrast agents, novel nanomaterials have drawn a lot of attention for therapy [12,13].

Among these, gold nanoparticles stand out as promising candidates for OCT contrast enhancement due to their bioinertness [14,15], customizable absorption and scattering cross-sections, and tunable surface plasmon resonance (SPR) peaks, which can be matched to specific OCT light sources. Furthermore, gold nanoparticles have significantly larger optical extinction cross-sections compared to organic dyes, providing a substantial advantage in terms of contrast enhancement and imaging efficiency [11,16]. Among gold nanoparticles, gold nanocages—structured particles with a hollow interior and a thin yet robust porous wall—have garnered attention. These particles are synthesized using a galvanic replacement reaction between Ag nanocubes and HAuCl_4_ in an aqueous solution. The SPR peaks of gold nanocages can be tuned to the near-infrared (NIR) region by adjusting the thickness and porosity of their walls [17]. Our previous work also highlights the biocompatibility and good size stability of gold nanocages in phosphate-buffered saline (PBS) and biological media [18]. Additionally, several other studies show that gold nanoparticles, ranging from 1.5 nM to 300 µM, exhibit no toxicity to cells [19,20,21]. Compared to gold nanorods and nanoshells, gold nanocages exhibit stronger absorption in the NIR region while maintaining a smaller size, which is crucial for effective delivery through tissues. These features give gold nanocages distinct advantages for use as contrast agents in OCT, as they provide a balanced combination of tunable optical properties and efficient delivery. Gold nanorods, for example, have a tunable SPR peak that typically occurs in the visible to NIR range, but their size, toxicity, and shape can limit their application [22,23]. Gold stars, another novel class of gold nanostructures, on the other hand, show unique structural properties, with multiple tips that offer an extended surface area, potentially enhancing interaction with biological tissues; however, their irregular shape and tendency to aggregate easily can make them challenging to deliver and manipulate in vivo [24,25]. In comparison, gold nanocages combine the tunable SPR properties of gold nanorods with the benefits of a unique hollow structure. This architecture provides a larger surface area for functionalization, which can be tailored to specific applications, while still preserving strong optical extinction properties. This makes gold nanocages particularly well-suited as contrast agents for OCT imaging. Our previous work demonstrates that gold nanocages enhanced OCT intensity and spectroscopic imaging contrast at wavelengths around 800 nm [10,11].

OCT imaging using light sources centered around 1060 nm has become very popular, as this wavelength offers deeper tissue penetration due to lower optical scattering and minimal dispersion compared with OCT at 800 nm, and lower absorption when compared with OCT at 1300 nm [26,27]. However, most gold nanoparticles have SPR wavelengths below 1000 nm, making them unsuitable as contrast agents for OCT imaging at this wavelength. Recent studies explore the use of Zinc oxide nanoparticles (ZnONP) and of core NaYF4:Ho^3+^/Yb^3+^ and core@shell NaYF4:Ho^3+^/Yb^3+^@NaGdF4 nanoparticles as contrast agents for OCT imaging at 1060 nm [28,29]. Despite promising advancements, Zinc oxide nanoparticles (ZnONP), typically around 100 nm with aggregation up to 355–400 nm, and NaYF4:Ho^3^⁺/Yb^3^⁺ or core@shell NaYF4:Ho^3^⁺/Yb^3^⁺@NaGdF4 nanoparticles, particularly those with a NaGdF4 shell, raise toxicity concerns due to the presence of gadolinium (Gd), a heavy metal, especially at higher doses or with prolonged exposure. Achieving uniform size and morphology can also be challenging. In contrast, gold nanocages stand out for their excellent scattering, optical tunability, biocompatibility, and ease of synthesis, making them very attractive for OCT imaging and other optically based applications. In this study, we investigate the use of gold nanocages with an SPR peak around 1060 nm to enhance OCT imaging contrast.

While gold nanocages with SPR peaks across a broad wavelength range can be synthesized following established protocols [17], it remains challenging to produce small gold nanocages with SPR peaks at longer wavelengths (i.e., beyond 950 nm). This challenge arises because the nanocages tend to collapse during the final stage of the galvanic replacement reaction, particularly when the concentration of HAuCl_4_ increases. In this work, we demonstrate that gold nanocages with SPR peaks shifting to approximately 1062 nm and beyond can be synthesized by reducing the reaction temperature and titration rate by 3–4 fold. We also quantitatively characterize the optical properties of these nanocages using an integrating sphere and the inverse adding–doubling method [30]. Finally, we present real-time OCT imaging with a 1060 nm swept source, performed on both tissue phantoms and in vivo mouse tissues, illustrating the contrast enhancement achieved with the gold nanocages.

## 2. Materials and Methods

### 2.1. Synthesis of Gold Nanocages with an SPR Wavelength Above 1000 nm

#### Materials

Sodium sulfide nonahydrate (Na_2_S·9H_2_O), ethylene glycol, and sodium chloride (NaCl) crystals were obtained from J.T. Baker (VWR, Radnor, PA, USA). Hydrogen tetrachloroaurate trihydrate (HAuCl_4_·3H_2_O), silver nitrate (AgNO_3_), polyvinylpyrrolidone (PVP), acetone, and ethanol were purchased from Sigma-Aldrich (St. Louis, MO, USA).

Gold nanocages with a longer SPR wavelength were synthesized following the galvanic replacement reaction principle as described by Skrabalak et al. [17], with a modified procedure. Briefly, 100 µL of an aqueous silver nanocube solution at a concentration of approximately 6 nM was first dispersed in 5 mL of deionized water containing poly(vinyl pyrrolidone) (1 mg/mL) in a 50 mL flask under magnetic stirring at room temperature (instead of the conventional ~90–100 °C) for approximately 10 min. Then, an aqueous solution of HAuCl_4_ (0.1 mM) was added to the flask using a two-channel syringe pump at a rate of 0.25 mL/min (about one-third of the conventional titration rate) while maintaining magnetic stirring. During the titration of the HAuCl_4_ solution, a series of color changes were observed, and the position of the SPR peaks of the gold nanocages was monitored using a UV-Vis-NIR spectrophotometer. The titration of the HAuCl_4_ solution was stopped once an appropriate SPR peak was reached. The reaction solution was then transferred to a 50 mL centrifuge tube and centrifuged at 5000 rpm for approximately 5 min. The supernatant was discarded, and the gold nanocages were redispersed in a saturated NaCl solution to dissolve and remove AgCl, which was generated during the galvanic replacement reaction. Finally, the solution was centrifuged at 10,000 rpm for about 10 min. The gold nanocages were thoroughly washed with deionized water and redispersed in 18.1 MΩ·cm E-pure water for further use.

### 2.2. Measurement of Optical Properties of Gold Nanocages

The optical properties of the gold nanocages were quantitatively characterized using the well-established integrating sphere method in conjunction with a spectrophotometer (Ocean Optics, Dundee, FL, USA). Measurements were conducted over the wavelength range of 1000 to 1200 nm. The nanocage solutions (~1 nM) in glass vials were ultrasonicated in a water bath for 20 min prior to measurement. Custom-made glass cuvettes with a thickness of 1 mm and dimensions of 25.5 × 25.5 mm were used to contain the solutions during the integrating sphere measurements.

### 2.3. OCT Imaging of Phantoms and In Vivo Mouse Models with Gold Nanocages as Contrast Agent

Swept-source OCT (SS-OCT) imaging was performed on phantoms and mouse tissues in vivo, both with and without the administration of gold nanocages. The as-synthesized gold nanocages had an edge length of approximately 65 nm, an SPR peak around 1040 nm, and a full-width-at-half-maximum (FWHM) spectrum bandwidth of ~300 nm, which overlaps with the 1060 nm OCT source spectrum. For phantom experiments, each phantom was prepared from 5% gelatin embedded with 1 mg/mL of Titania granules to mimic background tissue scattering. The scattering coefficient (µs) of the phantom was estimated to be approximately 2.65 mm^−1^, based on the decay curve of the OCT image. Gold nanocages were added to the phantom to a final concentration of 2 nM. SS-OCT imaging was conducted using a swept laser with a center wavelength of 1060 nm and a FWHM spectral bandwidth of ~65 nm.

For in vivo tissue experiments, a 50 µL aqueous solution of gold nanocages at a concentration of approximately 4 nM was locally injected into the tail tissue of an NCR nude mouse under anesthesia. While the phantom experiment provided a controlled environment where attenuation could be easily observed at a 2 nM concentration, the in vivo tissue (mouse tail) is more complex due to factors such as blood flow, tissue heterogeneity, and a more dynamic environment. A higher concentration of 4 nM gold nanocages was selected to ensure a more pronounced contrast enhancement in this in vivo system. OCT imaging was performed using the same method as described above. All animal experiments comply with the ARRIVE guidelines and were carried out in accordance with the National Institutes of Health Guide for the Care and Use of Laboratory Animals. All data reported in this study were collected at the Johns Hopkins University (JHU) with approval from JHU Animal Care and Use Committee (under the protocol of MO24M361).

## 3. Results

### 3.1. Gold Nanocage Synthesis, Influence of Synthesis Conditions, and Characterization

Figure 1 shows the UV-vis-NIR extinction spectra of the gold nanocages versus the titrated volume of the HAuCl_4_ solution to the reaction solution. The SPR peak continuously shifted towards a longer wavelength as the total volume of HAuCl_4_ increased, and an SPR peak wavelength of around 1062 was achieved after 5.2 mL of HAuCl_4_ solution was titrated. Figure 2 illustrates representative transmission electron microscopy (TEM) images of the nanostructures that were collected at different stages during the synthesis reaction. Figure 2A shows the silver nanocubes of an ~60 nm edge length which served as the templates for gold nanocage synthesis. After Ag nanocubes reacted with HAuCl_4_ in solution, a pinhole could be observed on the wall (Figure 2B); the SPR peak at this early reaction stage was around 622 nm. As more HAuCl_4_ solution was added, the interior of the Ag nanocube templates was continuously dissolved to yield a hollow nanobox through a combination of galvanic replacement and alloying between Ag and Au (Figure 2C), and the SPR peak continuously shifted to a longer wavelength (e.g., around 782 nm). At this stage, the wall thickness of the nanocages measured from the TEM images was about 10.7 ± 0.6 nm. In the later stage, gold nanocages, with a hollow interior and porous wall (see Figure 2D), were obtained through the dealloying of the nanoboxes’ walls [17] and the SPR peak reached a much longer wavelength (e.g., around 1062 nm). At this stage, the measured wall thickness was about 7.2 ± 0.4 nm.

To demonstrate the effect of the reaction temperature and titration rate, conventional synthesis conditions were used, i.e., at a temperature of ~90–100 °C and a HAuCl_4_ solution titration rate of 0.75 mL/min, while the rest of the reaction conditions remained the same. Under these conditions, dealloying caused the nanocages to collapse and form small gold fragments or clusters of irregular shapes, resulting in a second SPR peak around 550 nm when trying to shift the SPR peak beyond 930 nm. Figure 3A,C shows a UV-visible-NIR extinction spectrum and a TEM image of the collapsed gold nanocages, respectively. When we decreased the reaction temperature to room temperature and the titration rate to 0.25 mL/min, the reaction between silver nanocubes and HAuCl_4_ was found to be stable, resulting in gold nanocages with SPR peaks above 1000 nm. Figure 3B and Figure 3D, respectively, show a representative extinction spectrum and a TEM image of gold nanocages with an SPR peak at 1062 nm, which were synthesized under the modified conditions. From the TEM image the average edge length of the nanocages was found to be 65.2 ± 4.6 nm.

The optical properties (i.e., the absorption and reduced scattering coefficients) of the gold nanocages were measured using the integrating sphere method. According to discrete dipole approximation (DDA) calculations, the anisotropy factor (g) is about zero for gold nanocages; hence the scattering coefficient (µ_s_) is the same as the measured reduced scattering coefficient [µ_s_’ = (1 − g) µ_s_ = µ_s_]. The absorption (σ_abs_) and scattering (σ_sca_) cross-sections can then be obtained from the absorption (µ_a_) and scattering (µ_s_) coefficient according to σ_abs, sca_ = µ_a, s_/([C]⋅N_A_), where [C] is the concentration of gold nanocages and N_A_ is the Avogadro constant. The absorption (σ_abs_) and scattering (σ_sca_) cross-sections of a gold nanocage at 1060 nm were found to be (1.55 ± 0.38) × 10^−15^ m^2^ and (3.14 ± 0.09) × 10^−16^ m^2^, respectively, from which the ratio of σ_abs_ to σ_sca_ is calculated to about 4.9, suggesting the interaction of light with these nanocages are dominated by optical absorption.

### 3.2. Gold Nanocages as Contrast Agents for OCT Imaging at 1060 nm

Intensity-based SS-OCT imaging was performed on phantoms and mouse tail tissues in vivo, both with and without the administration of gold nanocages. Figure 4A shows the intensity-based SS-OCT image of a phantom tissue without gold nanocages. The phantom consists of a gelatin matrix embedded with Titania granules to mimic tissue-scattering properties. Figure 4B displays the same phantom but with the addition of gold nanocages (at a concentration of 2 nM). The presence of gold nanocages clearly enhances the contrast, as seen by the faster decay of the image intensity versus depth, indicating higher optical attenuation due to the added strong absorption by gold nanocages at the OCT imaging wavelength of 1060 nm. Figure 4C compares the representative decay curves of the backscattering intensity as a function of imaging depth for the phantom with and without gold nanocages. In the presence of nanocages, the curve exhibits a significantly faster (steeper) intensity decay slope, as expected from the enhanced contrast observed in Figure 4B. This demonstrates the effectiveness of gold nanocages as contrast agents in OCT imaging, by increasing signal decay rates. Quantitative analyses using methods described in our previous publications [10,31,32,33] revealed that the attenuation contrast increased from 2.65 mm⁻^1^ (in the phantom without gold nanocages) to 5.08 mm⁻^1^ (in the phantom containing 2 nM gold nanocages). The resulting change in the attenuation coefficient (2.43 mm⁻^1^) closely matches the expected value of 2.32 mm⁻^1^ for 2 nM gold nanocages, based on the measured attenuation cross-section described in the above section (σ_att_ = σ_abs_ + σ_sca_ = (1.86 ± 0.38) × 10⁻^15^ m^2^).

Figure 5A shows the intensity-based SS-OCT image of mouse tail tissue before the administration of gold nanocages. The image captures the complex structures the skin, with the imaging contrast dominated by the inherent scattering properties of the tissue. Figure 5B displays the same mouse tail tissue after the local injection of gold nanocages 50 uL of 4 nM in PBS). Similar to the phantom image (Figure 4B), OCT intensity exhibits faster decay along depth, indicative of higher attenuation primarily due to the local presence of gold nanocages. Quantitative analyses, using a method similar to that employed in the phantom imaging studies described above, revealed that the optical attenuation in skin tissue increased by approximately 3.05 mm⁻^1^ due to the presence of gold nanocages, corresponding to a gold nanocage concentration of 2.63 nM in tissue. This change in contrast represents the effect of gold nanocages with an SPR around 1060 nm as a contrast agent. The change in optical attenuation would be a very valuable parameter when performing quantitative OCT imaging [32,33,34].

## 4. Discussion

As demonstrated by the results shown in Figure 3, temperature and titration rate are critical parameters that control the reaction rate and allow for the tuning of the SPR peak wavelength to values above 1000 nm during the synthesis of gold nanocages. By lowering both the reaction temperature and titration rate compared to the conventional synthesis protocol for gold nanocages, we reduced the stress on the nanocage walls, thereby preventing collapse due to the stress corrosion cracking mechanism, as described in previous studies [35,36]. It is well-established that temperature has a significant effect on environmental cracking susceptibility, with crack velocity decreasing as the temperature drops, thus minimizing the apparent heat activation energies associated with the process [37]. Additionally, reducing the dealloying temperature significantly lowers the interfacial diffusivity of gold atoms, which leads to the formation of an ultrafine and stable nanoporous structure, thereby enhancing the chemical and physical properties of the nanocages [38,39]. Further reduction in the titration rate during the galvanic replacement reaction also helps reduce the potential stress on the walls of the nanocages, preventing collapse and promoting stability.

These findings provide important insights not only into the structural integrity of gold nanocages but also into how these synthesis parameters influence their optical properties. Building on this understanding, our study expands on the role of these critical parameters in controlling the optical characteristics of gold nanocages. While additional experimental data, such as pharmacokinetics or biodistribution studies, are needed for future research, the current results offer a more comprehensive understanding of the observed phenomena. Specifically, we highlight the potential advantages of gold nanocages for OCT imaging, particularly in applications requiring longer wavelengths, such as 1060 nm.

In this context, the synthesis conditions, particularly the reduced temperature and titration rate, play a crucial role in precisely tuning the SPR peak position. Under the reduced temperature and titration rate conditions, the SPR peak position could be precisely tuned to above 1000 nm by controlling the total volume of the HAuCl4 aqueous solution titrated into the reaction mixture, as shown in Figure 1. The SPR peak position of the gold nanocages is primarily determined by the wall thickness and porosity, which change during synthesis. As the porosity increases and the wall thickness decreases (as observed in Figure 2), the SPR peak shifts continuously to longer wavelengths. These observations are in agreement with our previous calculations based on discrete dipole approximation models for gold nanocages with an SPR peak around 800 nm.

Integrating sphere measurements demonstrated that the gold nanocages had a large extinction cross-section (~1.86 × 10^−15^ m^2^) at 1060 nm, which is difficult to achieve with organic dyes. Compared with typical organic dyes, such as Indocyanine Green (ICG), the extinction cross-section of the gold nanocages is approximately five orders of magnitude larger [40]. Furthermore, it was observed that the optical extinction at 1060 nm is dominated by absorption, as confirmed by the integrating sphere measurements. The strong absorption of gold nanocages results in significant optical attenuation when used as an absorbing imaging contrast agent. In highly scattering human tissues, these absorbing agents provide stronger change in contrast compared to scattering-based agents. In Figure 4B and Figure 5B, the presence of gold nanocages enhances tissue absorption, affecting the OCT intensity decay along depth which can be conveniently visualized. This enhanced optical absorption contrast is clear in the phantom and tissue OCT images (Figure 4B,C and Figure 5B). This suggests not only their potential for improving OCT imaging but also their broader applications, including photothermal therapy, which will be explored in future studies. Notably, this is the first demonstration of an OCT contrast agent using gold nanoparticles for imaging in the longer wavelength range (around 1060 nm). These findings align with our previous study, where gold nanocages were used as absorption-dominant contrast agents for OCT imaging at around 800 nm [10].

Building on these promising optical properties, the next step in understanding their practical applications involves refining the synthesis conditions and investigating their broader potential in clinical and research settings. Through these refinements, our results provide a clearer and more in-depth analysis of the synthesis conditions and optical properties, better aligning with the potential clinical and research applications of gold nanocages in OCT imaging. Further studies, particularly in vivo biodistribution and pharmacokinetics, will be essential to fully understand the clinical translation of these materials in medical imaging and therapy in future research.

## 5. Conclusions

In conclusion, we presented a modified protocol for synthesizing gold nanocages that shifts their SPR peak towards longer wavelengths, specifically above 1000 nm. These nanocages represent a novel class of contrast agents for intensity-based OCT imaging at wavelengths beyond 1000 nm. Experiments with tissue phantoms and in vivo mouse tissues, using a 1060 nm SS-OCT system, demonstrated enhanced absorption contrast with the as-synthesized gold nanocages. Additionally, the strong absorption properties of the gold nanocages suggest potential applications for photothermal therapy, although this aspect is beyond the scope of the current study [18].

## Figures and Tables

**Figure 1 nanomaterials-15-00755-f001:**
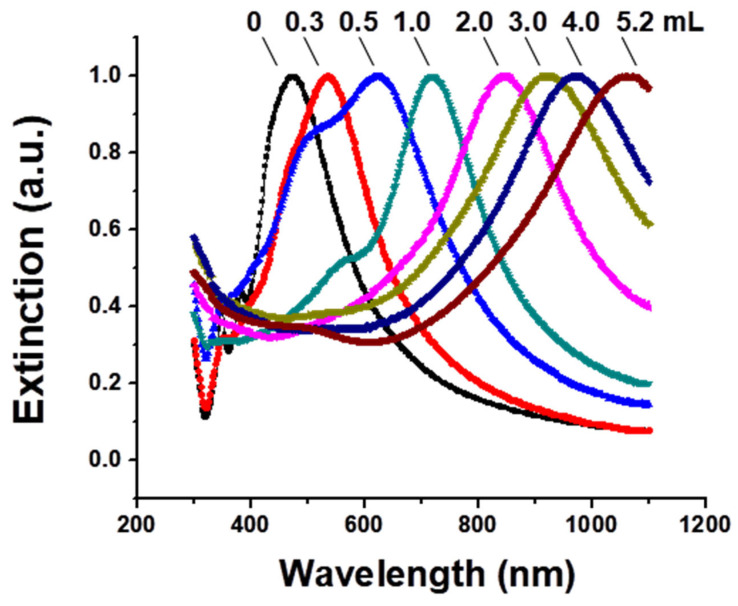
UV-vis-NIR extinction spectra of gold nanocages were measured as a function of the titrated volume of 0.1 mM HAuCl4 solution, using silver nanocubes as a template. The SPR peak shifted to longer wavelengths, reaching around 1062 nm as the HAuCl_4_ volume increased.

**Figure 2 nanomaterials-15-00755-f002:**
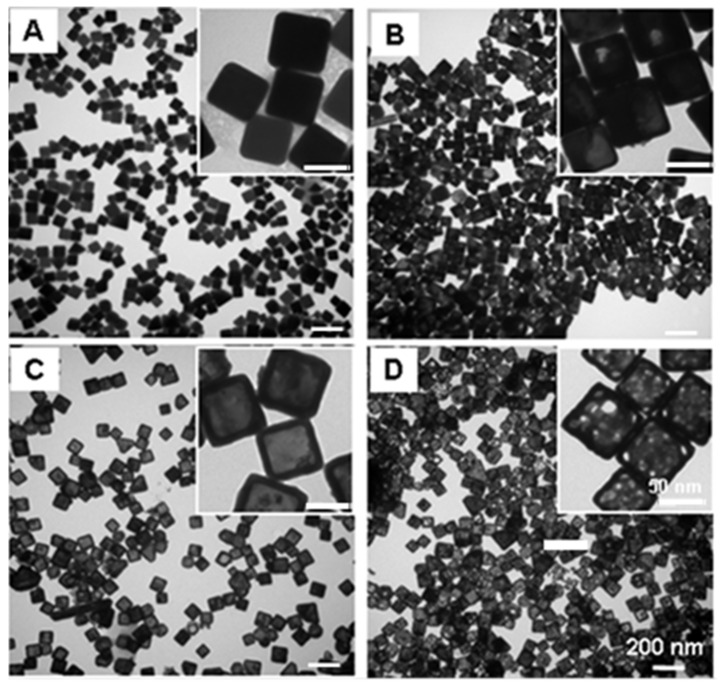
TEM images of (**A**) 60 nm Ag nanocubes which serve as the templates for gold nanocage synthesis; (**B**) Au/Ag alloy nanostructures with a pinhole observed on the wall during the early stage of the galvanic replacement reaction between the Ag nanocubes and HAuCl_4_ in aqueous solution; (**C**) Au/Ag alloy nanoboxes with an SPR peak around 782 nm; and (**D**) Au nanocages (after continuous dealloying) as the end product of the galvanic replacement reaction with a longer SPR peak around 1062 nm. Pores could be clearly observed on the wall of the nanocages. The insets show a zoomed-in TEM image of the Ag nanocubes, Ag/Au nanostructures, nanoboxes, and Au nanocages, respectively.

**Figure 3 nanomaterials-15-00755-f003:**
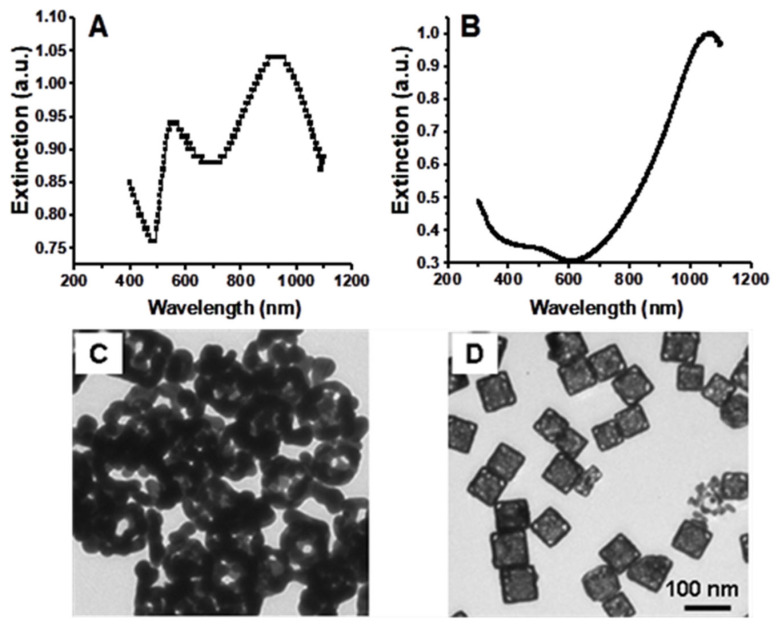
UV-vis-NIR extinction spectra (**top**) and TEM images (**bottom**) of gold nanocages. (**A**,**C**): under the conventional synthesis conditions (i.e., ~90–100 °C with a titration rate of ~0.75 mL/min), the reaction resulted in an SPR peak around 930 nm and a new peak around 550 nm. (**B**,**D**): under the modified synthesis conditions (i.e., room temperature with a reduced titration rate of ~0.25 mL/min), the reaction resulted in a single SPR peak around 1062 nm.

**Figure 4 nanomaterials-15-00755-f004:**
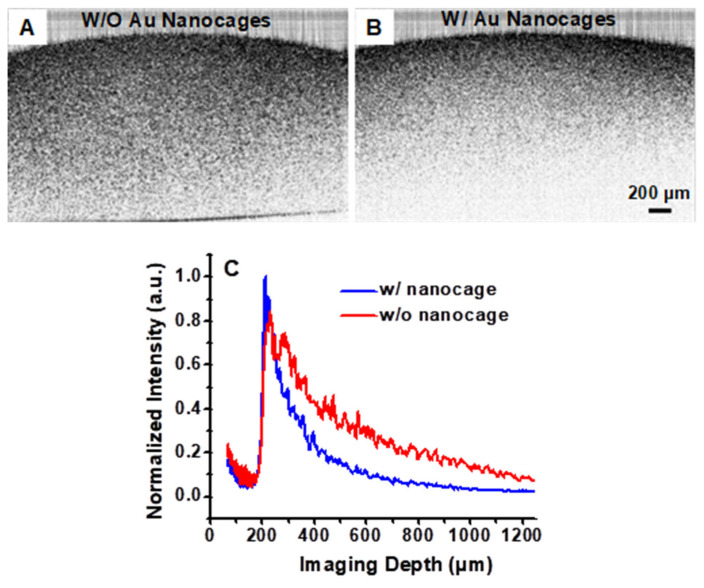
OCT images (obtained with a 1060 nm swept source) of a gelatin phantom embedded with TiO_2_ (1 mg/mL). (**A**,**B**) Intensity-based SS-OCT images of the phantom without (w/o) and with (w/) gold nanocages, respectively. (**C**) SS-OCT intensity signals on a linear scale as a function of imaging depth. Note that the SS-OCT signal recorded from the portion of the phantom with gold nanocages decays faster than that from the portion without nanocages.

**Figure 5 nanomaterials-15-00755-f005:**
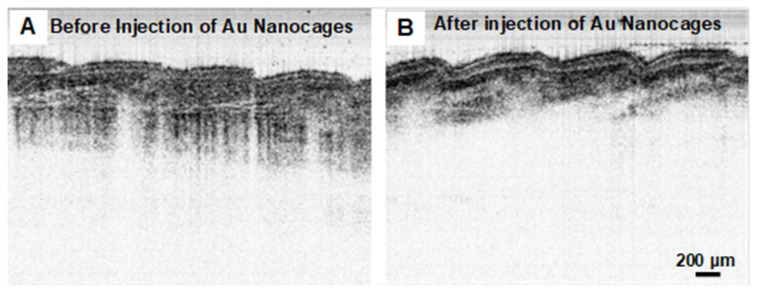
OCT images (with a 1060 nm swept source) of mouse tail tissue. (**A**,**B**) Intensity-based OCT images before and after the administration of nanocages (by local injection), respectively. The presence of gold nanocages increases the effective tissue absorption for intensity-based SS-OCT imaging.

## Data Availability

The original contributions presented in this study are included in the article. Further inquiries can be directed to the corresponding authors.

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
