# Peer review of "Gold Nanocages with a Long Surface Plasmon Resonance Peak Wavelength as Contrast Agents for Optical Coherence Tomography Imaging at 1060 nm"

_nanomaterials, 2025, doi:10.3390/nano15100755_

Round 1

Reviewer 1 Report

Comments and Suggestions for Authors

The author has reported an novel gold nanocages-based OCT agents with 65 nm size and 1060 nm SPR. Such nanocages demonstrated good performance on OCT imaging of mouse tail tissue. In summary, this study is well-organized and thoroughly documented, making it representative and meaningful. With minor revisions, this submission should be well-positioned for publication in Nanomaterials.

  1. Comparing with dyes, what are the main advantages of Gold nanoparticles for OCT imaging?
  2. How is the size stability of Au nanocages in PBS or medium?
  3. Before in vivo tests, the cell toxicity of Au nanocages on normal cell should be evaluated.
  4. 4. Recently literatures related to the metal-based agents for in vivo imaging should be also cited such as Commun., 2022, 13, 2009; Sci. China Chem., 2023, 66, 155; PNAS, 2022, 119, e2209904119; Anal. Chem., 2024, 96, 4495.

Author Response

Dear Reviewer,

Thank you for your valuable feedback. We agree with your comments and have attached our responses.

Reviewer 2 Report

Comments and Suggestions for Authors

In this study, the authors successfully synthesized gold nanocages with a small edge length (~65 nm) and characterized their optical properties, revealing a surface plasmon resonance peak around 1060 nm. Additionally, the potential of these nanocages as contrast agents for optical coherence tomography (OCT) at 1060 nm was explored. While the paper is well-organized, several major improvements are recommended before publication:

1. The introduction should be updated to comprehensively discuss the advantages of gold nanocages and compare them with other gold nanostructures, such as gold stars and gold nanorods. This will provide a clearer context for the significance of the current work.
2. Relevant literature, including studies such as https://doi.org/10.1002/bmm2.12123 and DOI: 10.1039/d2nr03186h, should be cited and discussed to strengthen the background and contextualize the findings.
3. Experiments assessing the biocompatibility of the nanocages, such as MTT assays, should be included to determine the optimal dose concentration for in vivo applications.
4. The imaging results require additional quantitative data to further analyze the efficacy of the nanocages as contrast agents. Time-dependent controls should also be provided to enhance the robustness of the conclusions.
5. The manuscript should incorporate recent publications from 2025 to ensure that the discussion reflects the latest advancements in the field. 

Author Response

Dear Reviewer,

Thank you very much for your valuable feedback. We agree with your comments and have attached our responses. Thanks

Reviewer 3 Report

Comments and Suggestions for Authors

Manuscript of Li and co-workers describes the synthesis and optical characterization of gold nanocages with a small edge length (~65 nm) and a surface plasmon resonance peak around 1060 nm. Manuscript is well written and experiments well designed and performed. However, in my opinion, the innovation of this paper is not sufficient to satisfy novelty criteria of Nanomaterials. Probably, the authors should better stress the novelty respect to the literature. In addition, some revisions are required:

  • some references on the properties of gold nanoparticles cited in lines 35-37 are required, in particular some references about the bioinertness and the non-toxic properties are mandatory; the same also about gold nanocages
  • line 51: authors reported that polipyrrole nanoparticles have been recently used as contrast agent for OTC imaging (ref 14), however this paper is published in 2011. it is not very recent!
  • in Figure 1 a wider range respect to those described in the main text is reported
  • in Figure 2, scale must be reported in all figures
  • in figure 4C, the meaning of w/ and w/o must be clarified
  • I believe that figure 5 needs to be better discussed. The authors must clarify the dark and white zone of tissue

Author Response

(The authors gave the same response as above.)

Reviewer 4 Report

Comments and Suggestions for Authors
  1. The manuscript should be edited and submitted by using the template of this journal.
  2. More details and presentations for Fig. 4 and 5 should be included.
  3. What is “w/” and “w/o” in the figures?
  4. The reference format should be double-checked and modified.
  5. Discussion Part can be integrated with Result Part.

Author Response

(The authors gave the same response as above.)

Reviewer 5 Report

Comments and Suggestions for Authors

The manuscript "Gold Nanocages with a Long SPR Peak Wavelength as Contrast 2
Agents for Optical Coherence Tomography Imaging at 1060 nm", written by Yongping Chen et al., describes a relatively interesting approach for the synthesis of gold nanocages for OCT. The theme per se is potentially strong and development of such contrast materials is demanding. The introduction part is written well and is based on current and topically relevant references. The Material part explains applied processes and reactions and is also well written. The Results part contain several gaps which need to be properly addressed. The Discussion part summarizes  the obtained results. Here, i would suggest to add more similar nanomaterials published before and compare their performance with the material prepared by the authors. For now, the Discussion part serves more as a Conclusions part, which is missing here. Regarding the results, here I strongly suggest to add more characterization techniques which would help to understand the physical characteristics of the prepared nanomaterials. First, i would suggest to add data from the measurements of zeta-potential for stability and surface chemistry and data from AFM to see the prepared material also in 3D space. It would be also beneficial to add DLS data to see the overall dispersity of the sample, which could also help to answer the hypothesis regarding the synthesis of nanocages at higher temperatures.

Moreover, data shown in figure 4 should be clarified as the only difference is the attenuation of the signal, which is not very clearly observable from the present OCT images. Here I suggest to use a better example to demonstrate the effect of the cubes on the obtained signal.

Author Response

Dear Reviewer,

Thank you so much for your valuable comments. Please find our responses in the attachment. 

Round 2

Reviewer 2 Report

Comments and Suggestions for Authors

Most of the key problems are just superficially answered by the authors, without any furher experimental support or additional data, thus leaving the problems unsolved. The current version can not be considered for publication.

Comments on the Quality of English Language

Could be improved.

Author Response

Dear reviewer,

Thank you for your valuable feedback. We have carefully revised our manuscript in this second round and addressed your comments. Please find our revised manuscript and detailed responses enclosed

Reviewer 3 Report

Comments and Suggestions for Authors

manuscript can be published in the present form

Author Response

Dear reviewer,

Thank you for agreeing that our manuscript is suitable for publication in its current form. Nevertheless, we have made further improvements to the manuscript during the second round.

Reviewer 4 Report

Comments and Suggestions for Authors

Accept in present form

Author Response

Dear reviewer, 

Thank you for accepting our manuscript as suitable for publication.

Reviewer 5 Report

Comments and Suggestions for Authors

The authors adequately improved the manuscript presented here in the revised version. My comments were suitably answered. I do not have additional comments.

Author Response

Dear reviewer,

Thank you for fully accepting our manuscript and for your kind comments: “The authors adequately improved the manuscript presented here in the revised version. My comments were suitably answered. I do not have additional comments.”

We greatly appreciate your time and thoughtful review.

Round 3

Reviewer 2 Report

Comments and Suggestions for Authors

1, Previously, Question:systematic studies evaluating biodistribution, optimal dosing, pharmacokinetics, and long-term toxicity are necessary for this work.

Response: We fully agree that systematic studies evaluating biodistribution, optimal dosing, pharmacokinetics, and long-term toxicity are crucial for translation. These important aspects will be the focus of our future work and fall beyond the scope of this initial feasibility report.

New Question: Although the reviewer could understand the authors' idea, systematic studies evaluating biodistribution, optimal dosing, pharmacokinetics, and long-term toxicity are really significant, as authors understand, then these data should be provided in this work, rather than in the future, because they are really matter to the current data.

2, Author response: Although we did not include new experimental data (such as additional pharmacokinetics or biodistribution data, which are critical for the next manuscript on the systematic administration of our nanocages), we have provided a more comprehensive discussion of the  issues raised to this revision. 

Qustion: The reviewer believes that new and related experimental data are necessary to confirm the conclusions of this paper, or it would be suspective of the conclusions of this paper.

3, Authors' response: Regarding time-dependent controls, we acknowledge their importance. This study focused on a single time-point analysis with local injection of the nanocages for proof of concept test in tissues in vivo. For pharmacokinetics and biodistribution, we believe that future studies could address this aspect more thoroughly by using systemic injection of the nanocages.

Question: since the authors could understand the significance of the time-point analysis, then please provide the data.

Comments on the Quality of English Language

Could be polished.